# Co-Located Tests, Better AI Code: How Test Syntax Structure Affects Foundation Model Code Generation

## Abstract

AI coding assistants increasingly generate code alongside tests. How developers structure test code, whether inline with the implementation or in separate blocks, has traditionally been a matter of testing philosophy. We investigate whether this choice affects AI code generation quality.

We conduct a large-scale empirical study (830+ generated files, 12 models, 3 providers) using SEGA, a three-dimensional evaluation framework measuring Determinism, Preservation, and Correctness. Comparing inline test syntax (Python doctests) against separated test syntax (Rust #[test] blocks) on a d-ary heap implementation, we find that: (1) inline tests yield near-perfect preservation (100%) and correctness (92–100%) across all models; (2) separated tests expose stark model-tier gaps (0–100% correctness) and independence between preservation and correctness; (3) model behavior evolves across generations, and notably one model breaks the test suppression pattern of its three predecessors; (4) mechanistic analysis on 7 open-source architectures (6 transformers and a gated-linear Recurrent Neural Network (RNN)) reveals inline test markers receive 2.8–4.4× stronger attention in 5/7 models, with causal validation via knockout and steering experiments; the co-location mechanism extends to a non-transformer architecture, suggesting the design recommendation is robust to future architectural shifts. In the Foundation Model era, test syntax structure is a software design concern: co-locating tests with implementation code produces measurably better AI-generated code.

## Keywords

AI-assisted development, code generation, test infrastructure, mechanistic interpretability, software design

## 1 Introduction

The Foundation Model (FM) era is reshaping software development. AI coding assistants (GitHub Copilot, Claude Code, Cursor, Augment Code, Kiro) now generate substantial portions of application code, including test code [26, 32]. As AI-powered development becomes the norm rather than the exception, architectural decisions that once affected only developer ergonomics now affect AI tool effectiveness. One such decision is how to structure test infrastructure: **inline test syntax** (tests co-located with the implementation, e.g., Python doctests embedded in docstrings via >>> markers) versus **separated test syntax** (tests in dedicated blocks or files, e.g., Rust #[test] functions in a mod tests {} block). This choice has traditionally been a matter of testing philosophy and team convention. But does it matter for AI code generation quality?

We ask: does the placement and syntax of test code affect the quality of AI-generated code? If so, test framework design, coding conventions, and Integrated Development Environment (IDE) tooling become first-class software design concerns for AI-powered

development. This is fundamentally a **software engineering question**: it asks how a design choice about test infrastructure interacts with the tools developers use to build software, with implications for test framework design, model selection, Continuous Integration/Continuous Deployment (CI/CD) pipeline configuration, and coding standards in AI-assisted teams.

*Research Questions.* **RQ1** (Quality Impact): Does inline test syntax produce higher-quality AI-generated code than separated test syntax, as measured across determinism, test preservation, and functional correctness? **RQ2** (Language vs. Syntax): Is the observed quality difference explained by programming language difficulty or by test syntax structure specifically? **RQ3** (Model Evolution): Is the effect stable across model families, tiers, and generations, or does model evolution change the picture? **RQ4** (Mechanism): What internal model mechanisms explain the observed differences? RQ1–RQ3 constitute the primary empirical contribution. RQ4 provides mechanistic supporting evidence.

*Contributions.* (1) **Empirical finding**: Test syntax structure measurably affects AI code generation quality—inline tests produce near-perfect results across all tested models; separated tests expose model-tier gaps and reveal that preservation and correctness are independent dimensions. (2) **Evaluation framework**: SEGA (Statistical Evidence for Generative Accuracy), a three-dimensional framework (Determinism × Preservation × Correctness) with quality-region visualization for rapid model assessment. (3) **Mechanistic evidence**: Attention pattern analysis across 7 open-source architectures (6 transformers and a gated-linear RNN) with causal validation (knockout and state knockout experiments) and intervention feasibility (attention steering and state steering), explaining *why* inline syntax produces better results. The non-transformer result demonstrates the co-location mechanism is not a transformer-specific artifact, strengthening the robustness of the design recommendation across architectural paradigms. (4) **Design guidelines**: Actionable recommendations for test framework design, model selection, and CI/CD pipeline design in AI-powered development.

Section 2 reviews background and related work. Section 3 describes the SEGA methodology and experimental design. Section 4 presents empirical results. Section 5 provides mechanistic evidence. Section 6 discusses implications, and Section 7 concludes.

## 2 Background & Related Work

### 2.1 Testing Practices and the Co-Location Spectrum

Software testing practices span a co-location spectrum, a concept rooted in test organization patterns [21, 33]:

- **Maximum co-location**: Python doctests, where test cases appear *inside* the docstring of the function they test, using >>> markers. The test is literally embedded in the function's documentation.

- **Same-file separation**: Rust #[test], where test functions appear in a dedicated mod tests {} block at the bottom of the same file. Structurally separated but co-located in the same compilation unit.
- **Separate-file separation**: pytest, JUnit, Go table-driven tests, where test code resides in separate files or directories, with no structural connection to the implementation beyond naming conventions.

This spectrum reflects different testing philosophies [3]. Python doctests [27] serve dual purposes as documentation and specification; Rust's #[test] culture [31] is deeply embedded in the language ecosystem (enabled by #[cfg(test)] conditional compilation). Research on test-production code co-evolution [39] shows these organizational choices have lasting consequences for software maintenance. Note that only ~9% of Python developers use doctest [14], while Rust's #[test] is pervasive. The comparison is therefore between two extremes of the co-location spectrum, not between two equally prevalent practices.

## 2.2 AI Coding Assistants in Professional Development

AI coding assistants have moved from novelty to standard practice [23, 26]. The current generation of tools increasingly relies on Foundation Models for code generation, with Claude models leading benchmarks for software engineering tasks (SWE-bench [15], SWE-1). How these assistants handle test code varies: some generate tests alongside implementation [5], others preserve prompt-provided tests, and some suppress or reinterpret test code [29, 38]. Understanding these behaviors is essential for teams adopting AI-assisted development workflows.

## 2.3 Large Language Model (LLM) Code Generation Evaluation

Existing benchmarks (HumanEval [6], MBPP (Mostly Basic Python Programming) [2], SWE-bench [15], LiveCodeBench [13]) evaluate the *correctness* of generated code against reference solutions or test suites. EvalPlus [18] demonstrated that many "passing" solutions fail on edge cases, while MultiPL-E [4] extended evaluation across 18 languages. Yet these benchmarks do not measure:

(a) Whether prompt-provided tests are **preserved** in the generated output;
(b) Whether preserved tests actually **pass** when executed;
(c) Whether preservation and correctness are **correlated**, or independent.

To our knowledge, no prior work specifically studies how test placement syntax, i.e., the structural relationship between test code and implementation code in the prompt, affects code generation quality. This paper addresses that gap.

## 2.4 Mechanistic Interpretability for Code Models

Mechanistic interpretability (MI) aims to understand model behavior by examining internal representations: attention patterns, hidden states, activation pathways [9, 11]. Attention analysis has

**Table 1: Experimental conditions**

| Experiment | Lang. | Tests | Prompt Type | Models |
|---|---|---|---|---|
| Baseline | Python | 50 | Model-generated | 9 |
| Directives | Python | 64 | Prompt-specified | 7 |
| Dir. v2d7 | Python | 73 | Refined prompt | 3 |
| Test-guided | Rust | 28 | Prompt-specified | 9 |

revealed how models process syntactic structures in natural language [7, 35], and activation patching provides causal validation of identified circuits [20, 40]. Recent work has applied MI to code models, revealing how they represent syntactic structures and variable references [16, 36].

We use MI as **supporting evidence** for our software engineering finding, not as a primary contribution. Specifically, we analyze attention patterns from test markers to function tokens to explain *why* inline test syntax produces better generation outcomes. This positions MI as a diagnostic tool for software engineering research, not as an end in itself.

## 3 Methodology

### 3.1 Study Design

*Task.* D-ary heap priority queue implementation, a well-known data structure with practical applications (e.g., Dijkstra's shortest path algorithm). This task is non-trivial: it requires 6+ public methods (insert, extract_min, decrease_priority, is_empty, len, peek), generic type parameters, heap property maintenance, and handling of edge cases (empty heap, single element, duplicate priorities, stress tests with 10,000+ elements). We deliberately chose a single meaningful task over many simple ones: simple tasks (string manipulation, basic arithmetic) would be trivial for production models and thus uninformative, while non-trivial tasks stress the capability boundary of the small open models used in our MI analysis (Section 5). The d-ary heap sits at the right complexity level: meaningful for production-grade models, on the edge for 3B–7B models. Cost constraints ($n$=50 runs × 12 models × multiple experiments) further motivate depth over breadth.

*Languages.* Python (inline doctests) and Rust (separated #[test] blocks). These represent two extremes of the test co-location spectrum:

- Python doctests appear *inside* the docstring of the function they test, providing maximum co-location.
- Rust #[test] functions appear in a separate mod tests {} block after the implementation, constituting structural separation within the same file.

We address the Rust difficulty confound upfront: the quality differences we observe are attributable to test syntax handling, not language difficulty. The evidence for this claim appears in Section 4.4.

*Experiments.* Table 1 summarizes the experimental conditions.

### 3.2 Models and Justification

**12 models across 3 providers**: 9 Claude variants spanning 3 tiers and multiple generations (Haiku 3, 3.5, 4.5; Sonnet 4, 4.5; Opus 4, 4.1,

**Table 2: SEGA evaluation dimensions**

| Dimension | Definition | Measurement |
|---|---|---|
| Determinism | Output stability across runs | % identical outputs at temp=0 |
| Preservation | Structural fidelity to prompt | % of prompt tests present in output |
| Correctness | Functional accuracy | % of individual tests passing |

**Table 3: Preservation × Correctness quality regions**

| Region | Pres. | Corr. | Interpretation |
|---|---|---|---|
| (+, +) | High | High | **Ideal**: preserves tests, they pass |
| (+, −) | High | Low | **Dangerous**: looks right, isn't |
| (−, +) | Low | High | **Safe but opaque**: correct, no tests |
| (−, −) | Low | Low | **Failing**: neither works |

4.5, 4.6), and 3 non-Claude models (Mistral Medium, Devstral-2512, EssentialAI RNJ-1) for cross-provider validation.

**Justification for Claude-centric selection**: Claude models lead SWE-bench/SWE-1 benchmarks for code generation and are the backbone of AI coding tools used in professional development. Studying 9 Claude variants across 3 tiers × 3+ generations provides longitudinal insight into the models that power actual AI-assisted development.

**Runs**: 50 per model/experiment combination (10 for some experiments). All runs at temperature=0, justified by our finding that temperature=0 does not guarantee determinism (Section 4.7), consistent with prior work on LLM non-determinism [22].

## 3.3 SEGA: Three-Dimensional Evaluation

We evaluate AI code generation along three orthogonal dimensions (Table 2):

Each dimension is measured independently using language-native tools (doctest.testmod() for Python, cargo test for Rust), without custom extraction scripts. We report a measurement correction in the Appendix where a custom script yielded 88% vs. the correct 100%, underscoring the importance of native tooling.

*Quality-Region Normalization.* Each percentage is mapped to $[-1, +1]$ via $v = (p - 50)/50$, where $p$ is the percentage (0% → −1, 50% → 0, 100% → +1).

The purpose is not precision. A model at $(+0.88, +1.00, +1.00)$ and one at $(+1.00, +1.00, +1.00)$ are both excellent. The purpose is **quality-region identification**: the quadrant (2D) or octant (3D) where a model falls provides immediate visual assessment of its fitness for AI-assisted development.

*2D Preservation × Correctness quadrants.* Table 3 defines four quality regions:

The 3D extension (adding Determinism) yields 8 octants, enabling richer characterization, for example distinguishing a deterministic ideal model from one that is non-deterministic but functionally invariant.

**Table 4: Baseline results: model-generated Python doctests**

| Model | Runs | Det. | File Pass | Corr. |
|---|---|---|---|---|
| Claude 3 Haiku | 50 | 94% | **100%** | **100%** |
| Claude Haiku 4.5 | 50 | 100% | **100%** | **100%** |
| Claude Sonnet 4 | 50 | 100% | **100%** | **100%** |
| Claude Sonnet 4.5 | 50 | 100% | **100%** | **100%** |
| Claude Opus 4.5 | 10 | 100% | **100%** | **100%** |
| Claude Opus 4.6 | 10 | 30% | **100%** | **100%** |
| Mistral Medium | 50 | 0% | **100%** | **100%** |
| Devstral-2512 | 50 | 52% | **100%** | **100%** |
| EssentialAI RNJ-1 | 50 | 100% | **100%** | **100%** |

**Table 5: Inline test results: prompt-provided Python doctests**

| Experiment | Tests | Models | Pres. | Corr. |
|---|---|---|---|---|
| Directives | 64 | 7 | 100%* | 92–97% |
| Dir. v2d7 | 73 | 3 | 100% | 98.6–99% |

## 4 Results: How Test Syntax Affects AI Code Generation

This section addresses RQ1–RQ3 through a systematic empirical comparison of inline and separated test syntax across 12 models.

### 4.1 Baseline: Establishing Model Capability

All 9 models achieve **100% preservation and 100% correctness** when generating their own Python doctests (baseline experiment, 50 doctests), as shown in Table 4.

This baseline is critical: it establishes that every model *can* produce a correct d-ary heap implementation. Any quality differences in subsequent experiments are attributable to test handling, not coding ability.

### 4.2 RQ1—Inline Tests (Python Doctests): Near-Perfect AI Performance

When models must preserve prompt-provided doctests, results are consistently strong (Table 5).
*Except Claude 3.5 Haiku, which strips all doctests (0% preservation, a regression from Haiku 3 and 4.5 which preserve at 100%).

*Key observations.*
(1) **Preservation is near-universal**: All models except Claude 3.5 Haiku preserve 100% of prompt-provided doctests, including directives (+SKIP, +ELLIPSIS, <BLANKLINE>).
(2) **Individual test correctness is high**: 92–99% across all models and experiments. The failures are narrow:
   - *Repr mismatch* (most common): Item(10, 100) vs. Item(number=10, cost=100). Models produce correct logic but different string representations.
   - *Directive scope*: one model fixes the repr mismatch but encounters a +SKIP directive scoping issue (1 failure per file).
   - These are formatting errors, not logic errors.
(3) **File-level pass rate is misleading**: 2–3 failures out of 64 tests causes a file-level FAIL, but individual correctness is 95–97%. Evaluation granularity matters.

**Table 6: Separated test results: Rust `#[test]` blocks**

| Model Tier | Pres. | Compiles? | Pass | Behavior |
|---|---|---|---|---|
| Haiku 3 | 0% | No | 0% | Invalid Rust |
| Haiku 4.5 | 100% | Yes | 100% | Full success |
| Sonnet 4, 4.5 | 100% | Yes | 100% | Full success |
| Opus 4, 4.1, 4.5 | **0%** | Yes | N/A* | Suppresses tests |
| Opus 4.6 | **100%** | Yes | **100%** | Breaks pattern |
| Mistral Medium | 0% | Var. | Var. | Suppresses tests |

(4) **The preservation-correctness gap is narrow**: In Python, high preservation reliably accompanies high correctness. Inline test syntax is a strong signal that AI models preserve and implement correctly.

## 4.3 Separated Tests (Rust `#[test]`): Stark Model-Tier Gaps

When models must preserve prompt-provided `#[test]` blocks (28 tests including `#[ignore]` and `#[should_panic]`), results diverge sharply by model tier (Table 6).

*Opus 4/4.1/4.5 generate correct, compiling Rust implementations: the code works. But they suppress all 28 `#[test]` functions entirely, treating them as a specification to implement against rather than content to reproduce.

**Multi-run validation** (Opus 4.6, 10 runs): Despite producing 6 different code variants (30% determinism), every variant preserves all 28 tests, compiles without errors, and passes all 28 tests. Non-determinism is purely cosmetic (variable names, code structure, comments), not functional.

**The preservation-correctness gap is wide in Rust**: Opus 4/4.1/4.5 achieve 0% preservation but 100% correctness. Correct code, but no tests to verify it. This is the starkest evidence that preservation and correctness are **independent** dimensions.

## 4.4 RQ2—Language vs. Syntax

The quality difference is attributable to **test syntax handling**, not language difficulty. The evidence:
(a) **Opus 4/4.1/4.5 can write Rust**: They generate correct, compiling implementations. Their issue is test *syntax* suppression, not Rust *language* capability.
(b) **Tier inversion**: Haiku 4.5 (a lower-tier model) achieves 100% preservation AND 100% correctness in Rust, while Opus 4/4.1/4.5 (higher-tier models) suppress all tests. If the issue were language difficulty, higher-tier models would perform better, not worse at preservation.
(c) **Mechanistic evidence** (Section 5): The attention analysis shows the effect operates at the test *marker* level (>>> vs `#[test]`), not at the language grammar level.

## 4.5 SEGA Quality-Region Analysis

Mapping models to the 2D Preservation × Correctness space (normalized to [−1, +1]) reveals a striking contrast:

**Table 7: SEGA quality regions for Rust experiments**

| Model | Pres. | Corr. | Region |
|---|---|---|---|
| Sonnet 4/4.5, Haiku 4.5, Opus 4.6 | +1.00 | +1.00 | (+, +) Ideal |
| Opus 4, 4.1, 4.5 | −1.00 | +1.00 | (−, +) Opaque |
| Haiku 3 | −1.00 | −1.00 | (−, −) Failing |
| Mistral Medium | −1.00 | Variable | (−, −) to (−, +) |

**Python (all experiments)**: All models cluster in the (+, +) quadrant, with high preservation and high correctness. The discriminative power of Python doctests is low: all models look approximately the same.

**Rust (test-guided)**: Models spread across 3 of 4 quadrants (Table 7):

**Visualization insight**: The same models that are indistinguishable in Python are clearly separated in Rust. **Test syntax choice determines whether your evaluation can discriminate model quality.**

*3D extension (adding Determinism).* Opus 4.6: (−0.40, +1.00, +1.00) in Rust, non-deterministic but functionally perfect. Mistral Medium: (−1.00, +1.00, +1.00) in Python baseline, zero determinism (50 unique outputs) but perfect quality. Claude Sonnet 4: (+1.00, +1.00, +1.00) in both languages, the "perfect corner."

## 4.6 RQ3—Model Evolution: Behavioral Patterns Are Not Fixed

**The test suppression pattern and its breaking**: Three consecutive Opus generations (4, 4.1, 4.5) exhibit identical behavior on Rust `#[test]` blocks: they suppress all test functions, treating them as specification rather than content to preserve. **Opus 4.6 breaks this pattern**, preserving all 28 tests with 100% correctness. This is a training-level behavioral change; all four versions can write correct Rust, and the change is in how they handle test syntax.

**The Haiku regression**: Haiku 3 preserves Python doctests at 100%. Haiku 3.5 strips *all* doctests (0% preservation). Haiku 4.5 restores preservation to 100%. Model updates can introduce regressions in test handling behavior, even within the same tier.

**Implications**: Teams using AI assistants must **monitor model behavior across updates**. A CI/CD pipeline configured for one model may produce different results when the organization upgrades, not because code quality decreases, but because test handling behavior changes.

## 4.7 Methodological Finding: Temperature ≠ Determinism

Setting temperature=0 does **not** guarantee deterministic output [22] (Table 8).

Critically, **non-determinism is orthogonal to quality**: Opus 4.6 produces 6–11 unique code variants per experiment, yet *every variant* achieves 100% preservation and 100% correctness. The variation is cosmetic (variable names, code organization, comments), not functional.

**Table 8: Determinism at temperature=0**

| Model | Determinism at temp=0 |
|---|---|
| Mistral Medium | 0% (50 unique in 50 runs) |
| Devstral-2512 | 52% |
| Claude 3 Haiku | ~94% |
| Claude Opus 4.6 | 30–64% (experiment-dependent) |
| Claude Sonnet 4 | 100% |
| Claude Haiku 4.5 | 100% |

**Implication**: Evaluation pipelines for AI-generated code must use multi-run designs even at temperature=0. Single-run evaluations may not be reproducible.

## 5 Why Test Syntax Matters: Mechanistic Evidence

The empirical results (Section 4) establish *that* inline test syntax produces better AI code generation outcomes. This section addresses **RQ4** by investigating *why*, using mechanistic interpretability (MI) as supporting evidence for the software design finding. We analyze internal representations across 7 open-source models—6 transformers and a gated-linear Recurrent Neural Network (RNN)—to identify the mechanisms linking test syntax structure to generation quality and to test whether the co-location advantage is specific to transformer attention or generalizes across architectural paradigms.

### 5.1 Attention Pattern Analysis

*Setup.* We use a Rust-based toolkit for mechanistic analysis to measure how strongly test markers (>>> for Python, #[test] for Rust) attend to function signature tokens (def/fn, function name, parameters) across model layers. For transformers, we extract post-softmax attention weights directly. For RWKV-6 (a gated-linear RNN with no attention matrices), we compute effective attention from the Weighted Key-Value (WKV) recurrence.

*Models.* 7 open-source models spanning diverse architectures: Qwen2.5-Coder-7B and -3B [12], StarCoder2-3B [19], CodeGemma-7B [8], Code-LLaMA-7B [28], Phi-3-mini-4k-instruct [1] (6 transformers [34]), and RWKV-6-Finch-1B6 [25] (a gated-linear RNN [24]). MI requires access to internal representations that proprietary models do not expose. The 7 models were selected for architectural diversity (including a non-transformer paradigm), code competence, and feasibility on consumer hardware (16GB video RAM (VRAM)).

*Corpus.* 10 Python doctest samples and 10 Rust test samples, using a model-agnostic corpus format with character-level byte offsets (achieving 100% position accuracy across all tokenizer architectures without model-specific preprocessing).

*Key result.* In **5 of 7 models**, Python >>> markers show **2.8–4.4× stronger attention** to function tokens than Rust #[test] attributes ($p < 0.0002$, Welch's $t$-test [37]), as shown in Table 9.

For the 6 transformer models, we measure post-softmax attention weights directly. For RWKV-6 (a gated-linear RNN with no attention matrices), we compute *effective attention* from the WKV recurrence, producing comparable [batch, heads, seq, seq] matrices

via ReLU+L1 normalization of signed $r \cdot k$ products with cumulative decay. RWKV-6 achieves the **strongest statistical significance** of all 7 models ($t = 11.57$, $p < 10^{-7}$), with 23 of 24 layers showing $p < 0.05$.

The attention advantage is **not universal**: Phi-3-mini shows near-symmetric attention, and Code-LLaMA shows a reversed pattern (Rust > Python at all scanned layers). The effect is **architecture-dependent**, likely influenced by training data composition and pre-training objectives. Notably, RWKV-6 is a general-purpose model (not code-specialized), yet shows the effect more strongly than any code-specialized transformer, complicating a simple "code-specialization is necessary" narrative.

*Interpretation for software design.* Inline co-location, where test markers are physically adjacent to function signatures in the token stream, creates stronger semantic binding during generation. This is a **structural property of the test syntax**, not a property of the programming language. The >>> marker appears inside the docstring, immediately following the function it tests. The #[test] attribute appears in a separate block, creating greater token-stream distance. Crucially, this effect extends from transformer attention to RNN recurrent state dynamics, suggesting the co-location advantage is a fundamental property of sequence processing. This makes the design recommendation—co-locate tests with implementation code—robust to future architectural shifts.

### 5.2 Causal Validation: Knockout Experiments

Correlation does not imply causation. We conducted knockout experiments—removing attention from test markers to function tokens (transformers) or suppressing recurrent state writes at marker positions (RWKV-6)—and measured the impact on model predictions via Kullback-Leibler divergence [17], following activation patching best practices [40], to establish whether these pathways are causally important (Table 10).

**RWKV-6 state knockout**: RWKV-6 has no attention matrix to knock out. Instead, we suppress the kv write to recurrent state at marker positions (setting state = decay · state instead of state = $kv$ + decay · state), making the marker position invisible to all future tokens—semantically equivalent to transformer "all-edge" knockout. At layer 14, Python KL = 0.000362, Rust KL = 0.000177 (ratio 2.05×, $p = 0.060$). While marginally significant, the direction is consistent with the transformer finding: Python markers carry more causal weight than Rust markers.

*Findings.*

(1) **Causality confirmed for Qwen models**: The causal effect ratio (1878×) is ~500× larger than the correlational ratio (3.5×), demonstrating that small attention differences produce large causal differences.

(2) **Architecture-specific mechanisms**: Different architectures process test markers through fundamentally different pathways: Qwen processes Python via attention, Rust via non-attention pathways; StarCoder2 shows the reverse; Phi-3 and Code-LLaMA show near-zero effects for both.

**Table 9: Attention from test markers to function tokens across 7 architectures**

| Model | Architecture | Best Layer | Python $\mu$ | Rust $\mu$ | Ratio | $p$-value |
|---|---|---|---|---|---|---|
| Qwen2.5-Coder-7B | Transformer | 16 | 9.08% | 2.59% | 3.51× | 0.000003 |
| Qwen2.5-Coder-3B | Transformer | 14 | 8.47% | 3.05% | 2.78× | 0.000009 |
| StarCoder2-3B | Transformer | 23 | 7.19% | 2.41% | 2.98× | 0.000004 |
| CodeGemma-7B | Transformer | 24 | 5.23% | 1.20% | 4.35× | 0.000114 |
| **RWKV-6-Finch-1B6** | **Gated-linear RNN** | **14** | **6.43%** | **2.17%** | **2.96×** | $\mathbf{6.8 \times 10^{-8}}$ |
| Phi-3-mini | Transformer | 14 | 17.30% | 14.03% | 1.23× | 0.146 (n.s.) |
| Code-LLaMA-7B | Transformer | 26 | 9.71% | 12.23% | 0.79× | 0.188 (n.s.) |

**Table 10: Knockout experiment results (KL divergence)**

| Model | Py KL | Rust KL | Ratio | Pattern |
|---|---|---|---|---|
| Qwen-7B (L1) | 6.76% | 0.004% | 1878× | Strong Py |
| Qwen-3B (L0) | 153.7% | 0.19% | 806× | Extreme Py |
| StarCoder2 (L0) | 0.04% | 2.81% | 0.01× | Reversed |
| CodeGemma (L4) | 1.46% | 0.013% | 113× | Py bias |
| Phi-3-mini (L4) | 0.017% | 0.007% | 2.4× | Near-zero |
| Code-LLaMA (L8) | 0.00006% | 0.000006% | 10× | Near-zero |

**Table 11: Steering results: test preservation on 5 initial prompts**

| Model | Baseline | Steered | Change |
|---|---|---|---|
| Qwen-3B | 0/5 | 2/5 | **+40%** |
| Qwen-7B | 1/5 | 2/5 | +20% |
| StarCoder2-3B | 0/5 | 0/5 | No effect |
| CodeGemma-7B | 0/5 | 0/5 | No effect |
| Phi-3-mini | 0/5 | 0/5 | No effect |
| Code-LLaMA-7B | 0/5 | 0/5 | No effect |

(3) **Cross-paradigm consistency**: The Python > Rust causal asymmetry holds across both architectural paradigms (transformer attention and RNN state dynamics), suggesting the co-location advantage is not an artifact of the attention mechanism.

(4) **Practical implication**: Interventions to improve test handling must be architecture-specific. The knockout analysis can serve as a **diagnostic** for predicting which architectures will respond to a given intervention.

## 5.3 Proof-of-Concept: Attention and State Steering

Can we improve separated test syntax handling by boosting its attention to inline test levels? And does the knockout-predicts-steering relationship extend beyond transformers?

*5.3.1 Transformer Attention Steering.* **Approach**: Post-softmax attention steering, applying a scale factor to #[test] → function token attention weights, then renormalizing to preserve valid probability distributions.

**Safety**: KL divergence between steered and baseline outputs remains flat across all 6 transformer models and all steering intensities (0.5× through 9×), confirming safe intervention without catastrophic output changes.

**End-to-end generation results**: We tested steering on 50 diverse Rust code completion prompts per model. Initial results (5 prompts) showed improvement on Qwen (Table 11):

**Steering works where knockout predicts it should**: Qwen models, which process Python via attention and Rust via non-attention pathways, respond to steering. Models that already use attention for Rust (StarCoder2) or use non-attention pathways for both (Phi-3, Code-LLaMA) do not respond. **Knockout experiments reliably predict steering effectiveness across all 6 tested transformer architectures.**

**Hard discovery, model size as a bottleneck**: Scaling to $n$=50 with diverse prompts revealed a fundamental limitation: 3B and 7B parameter models are too small to generate correct Rust code for most prompt types. **Only prompts involving string-based functions** (simple types, no ownership/lifetime complexity) produced valid Rust output during steered generation.

*5.3.2 RWKV-6 State Steering.* For the non-transformer architecture, we extend steering from attention weights to recurrent state dynamics: scaling the kv write to recurrent state at marker positions (state = scale · $kv$ + decay · state, where scale=1.0 is identity and scale=0.0 is knockout). A dose-response experiment across 6 scale factors (0.0 to 9.0) at layer 14 reveals three findings relevant to the design recommendation's robustness:

(1) **The co-location asymmetry is causal in RNN state**: Python markers consistently produce higher KL divergence than Rust markers across all scales ($p < 0.05$ at 3 of 6 scales), confirming the knockout finding via an independent methodology.

(2) **Graded dose-response**: Unlike transformer attention steering (flat KL across intensities due to softmax normalization), RNN state steering produces a monotonic, graded response—Python KL increases 13× from dampened (scale=0.5) to amplified (scale=9.0). The recurrent state is an unbounded accumulator, making the intervention proportional rather than saturating.

(3) **Knockout-predicts-steering validated across paradigms**: The state knockout asymmetry (Python > Rust) correctly predicts state steering effects, extending the knockout-predicts-steering diagnostic to all 7 tested architectures—6 transformers and 1 RNN.

*5.3.3 Implications for Recommendation Robustness.* The critical software engineering question is not whether RWKV-6 generates

better code with inline tests—at 1.6B parameters, it cannot generate meaningful code. The question is whether the co-location mechanism observed in transformers is a **transformer-specific artifact** that could break with the next architectural shift, or a **general property of sequence processing** that makes the design recommendation durable.

The evidence supports the latter. The co-location asymmetry (Python markers binding more strongly than Rust markers) manifests through fundamentally different mechanisms—softmax attention weights in transformers, recurrent state dynamics in RWKV-6—yet produces consistent direction and statistical significance across both paradigms. The knockout-predicts-steering diagnostic, validated across all 7 architectures, provides a practical tool for assessing whether the recommendation applies to future architectures as they emerge.

**Practical constraint**: The models amenable to consumer-hardware MI (1.6B–7B) are too small for reliable Rust code generation, creating a gap between where we can *diagnose* and where interventions would *matter*. This diagnostic-deployment gap is inherent to accessible MI research.

## 5.4 Accessibility: MI for the Rest of Us

A deliberate design goal of this work was to demonstrate that meaningful MI research is achievable on hardware most researchers can afford, not just institutions with H100 clusters.

**Hardware**: All mechanistic experiments ran on a consumer Graphics Processing Unit (GPU) (16GB VRAM, ~$500). Early experiments on 8GB cards failed: 7B models exceeded memory during attention extraction. 16GB is the minimum viable threshold for MI on code LLMs in the 3B–7B range.

**Tooling choice**: We built the analysis toolkit in Rust using the candle ML framework rather than Python/PyTorch. This was motivated by VRAM efficiency: Rust's zero-cost abstractions and candle's minimal runtime overhead allow fine-grained memory control (KV-cache management, layer-by-layer processing, shared mask caching) that would be difficult to achieve in Python. The result is a toolkit that runs 7-model MI experiments (attention extraction, knockout, steering, generation) across two architectural paradigms within a 16GB envelope.

**Model selection rationale**: The 7 open-source models were selected at the intersection of three constraints: (a) diverse architectures for generalizability, explicitly including a non-transformer paradigm; (b) demonstrated code competence; (c) fits within 16GB VRAM with candle compatibility. RWKV-6 (1.6B parameters, ~3.2GB) was added specifically to test whether attention-based findings generalize to non-transformer architectures. The resulting findings ($p < 0.0002$ in 5/7 models including a non-transformer, architecture-dependent patterns, knockout-predicts-steering validation across both paradigms) demonstrate that statistically robust MI research is achievable within these constraints.

**Limits of accessibility**: The steering experiments reveal an inherent tension: models that fit on consumer hardware are large enough for attention and state analysis but too small for reliable Rust code generation. MI for the rest of us works for diagnosis; deployment-scale validation requires larger models and larger hardware.

The toolkit will be released as open-source upon acceptance.

## 6 Discussion

### 6.1 Test Syntax Design for the FM Era

Our results suggest a design principle for AI-powered development: **co-locate tests with the implementation code they verify**.

This is not merely "use Python doctests." The principle is **co-location**, placing test specifications in close structural proximity to the code they test within the token stream. This principle can be applied across languages and frameworks:

- **For test framework designers**: New frameworks should consider "AI-friendliness" as a design criterion. Prior work on prompt structure [41] shows that how context is presented matters as much as what context is provided. Inline or doc-adjacent test syntax creates stronger signals for AI code generation. Test frameworks that structurally co-locate test specifications with the functions they test are likely to interact better with FM-based code generators.
- **For IDE and tool designers**: Even if tests are stored in separate files (as with pytest or JUnit), IDE features that *present* tests inline during AI-assisted generation—for example injecting test specifications into the prompt context adjacent to the relevant function—could capture some of the inline attention benefit.
- **Caveat on doctest prevalence**: Only ~9% of Python developers use doctests [14]. Our recommendation is about the structural co-location principle, not about switching to a specific testing framework. The attention analysis confirms the relevant factor is token-stream proximity of test markers to function signatures, which is achievable through tooling regardless of the underlying test framework.

### 6.2 Model Selection for Typed Languages

Rust exposes a clear model-tier divide that Python hides. Haiku 3 cannot generate compilable Rust (0% correctness); Haiku 4.5 and all Sonnet variants achieve 100%. For software teams working in statically-typed languages with strict compilation requirements, **model tier is a procurement decision** that directly affects AI assistant effectiveness.

### 6.3 Trustworthiness of AI-Generated Code

Our results reveal two distinct dimensions of trustworthiness that are often conflated:

(1) **"Preserved" ≠ "Correct"**: Opus 4/4.1/4.5 achieve 100% correctness with 0% preservation in Rust: the code works, but there are no tests in the output to verify it. Conversely, Python experiments show 100% preservation with <100% correctness (repr mismatches). Preservation and correctness are **independent** dimensions that must be measured separately.

(2) **Whole-file ≠ individual test granularity**: In Python directives experiments, file-level pass rate is 0–2% while individual test correctness is 92–97%. A CI/CD pipeline that reports only file-level pass/fail dramatically understates the actual code quality.

**CI/CD recommendation**: Pipelines for AI-generated code must:
- **Run** generated tests, not just check they exist (preservation ≠ correctness)

- Use **language-native test runners** as ground truth, e.g., Python's `doctest.testmod()` or Rust's `cargo test`
- Report at **both file and individual test granularity**

## 6.4 Model Evolution and Software Engineering Practice

Model behavior is not fixed across versions. The Opus suppression pattern (3 consecutive generations: 4, 4.1, 4.5) was broken by Opus 4.6. The Haiku regression (3.5 strips doctests; 3 and 4.5 preserve them) shows that updates within a tier can introduce regressions.

**Practical recommendations**:
- **Version-pin models** in CI/CD pipelines; re-evaluate on model updates.
- **Tier predicts behavior but is not permanent**: use tier as a heuristic, but verify on your specific codebase.
- **Non-determinism requires multi-run evaluation**: even at temperature=0, some models produce 6–11 variants per experiment.

## 6.5 Threats to Validity

**Single task domain.** The d-ary heap priority queue is a single task, limiting generalizability—studies of LLM code bugs [10, 30] show failure patterns vary by domain. This was a deliberate trade-off: simple tasks (string manipulation, basic arithmetic) would be trivial for production models and thus uninformative, while non-trivial tasks stress the capability boundary of the small open models used in our MI analysis. The d-ary heap is a well-known data structure with practical applications (e.g., Dijkstra's algorithm), requires 6+ methods with generics and edge cases, and is complex enough to discriminate model capability across tiers. This choice is corroborated by the steering experiments (Section 5): when we scaled to $n$=50 diverse prompts, 3B–7B models could only generate valid Rust for string-based functions. Additionally: (a) the MI evidence shows the effect operates at the test *marker* level (not task content), suggesting the syntax effect is task-independent; (b) the structural nature of the finding (co-location vs. separation) is inherently task-agnostic. Future work should validate across diverse task domains.

**Claude-heavy model selection.** 9 of 12 models are Claude variants. This is mitigated by: (a) Claude leads SWE-bench/SWE-1 benchmarks for code generation; (b) AI coding tools increasingly rely exclusively on Claude, meaning we study the models powering actual AI-assisted development; (c) the 9 variants span 3 tiers × 3+ generations, providing longitudinal depth; (d) 3 non-Claude models provide cross-provider validation; (e) the MI analysis covers 7 diverse open-source architectures including a non-transformer.

**Rust difficulty confound.** Addressed in Section 4.4. Opus 4/4.1/4.5 compile correct Rust but suppress tests: the issue is syntax handling, not language capability. The MI analysis shows the effect at the marker level, not the language grammar level.

**Temperature=0 only.** By design, to enable reproducibility and to study determinism as a dimension. Limits generalization to typical usage where temperature > 0.

**MI on open-source models (1.6B–7B).** Proprietary models do not expose internal representations, making open-source models necessary for MI. Our model selection reflects a deliberate "MI for the rest of us" design: we constrained ourselves to models that run on a consumer GPU (16GB VRAM, ~$500) using Rust/candle for VRAM efficiency. Findings are statistically significant in 5/7 models; the 2 non-significant results (Phi-3, Code-LLaMA) are informative rather than null, as they reveal architecture-dependent processing. Whether the patterns observed in 1.6B–7B models transfer to 70B+ production models remains an open question.

**Steering generation limited by model size.** We scaled the transformer steering evaluation from 5 to 50 diverse prompts per model. This revealed a fundamental limit: 3B–7B models are too small for reliable Rust code generation beyond simple function signatures. The knockout-predicts-steering methodology is validated across all 7 architectures (the key methodological contribution), but the end-to-end generation evidence is necessarily limited to prompt types these small models can handle.

## 7 Conclusion

We investigate whether test syntax structure, inline versus separated, affects AI code generation quality. Through a large-scale empirical study (830+ generated files, 12 models, 3 providers) and mechanistic analysis (7 open-source architectures spanning transformers and a gated-linear RNN), we find:

**RQ1**: Inline test syntax (Python doctests) produces near-perfect AI generation (100% preservation, 92–100% correctness) across all models. Separated test syntax (Rust `#[test]`) exposes stark model-tier gaps (0–100% on both dimensions) and reveals preservation and correctness as independent dimensions.

**RQ2**: The quality difference is primarily attributable to test syntax structure, not language difficulty. Opus models write correct Rust but suppress test syntax; lower-tier Haiku 4.5 preserves it.

**RQ3**: The effect varies by model tier and evolves across generations. The Opus test suppression pattern (4/4.1/4.5) was broken by Opus 4.6. Teams must monitor model behavior across updates.

**RQ4**: In 5/7 architectures (including a non-transformer gated-linear RNN), inline test markers receive 2.8–4.4× stronger attention to function tokens ($p < 0.0002$; RWKV-6: $p < 10^{-7}$). Knockout experiments confirm causality across both architectural paradigms. Steering improves test preservation where knockout predicts (+40% on Qwen). The co-location mechanism extends to non-transformer state dynamics with validated dose-response, suggesting the design recommendation is robust to future architectural shifts.

**Design guidelines for AI-powered development**:
(1) **Co-locate tests with implementation code** when developing with AI assistants.
(2) **Run generated tests; don't trust preservation** as a proxy for correctness.
(3) **Evaluate model behavior on your specific language/syntax combination**; don't assume cross-language generalization.

In the Foundation Model era, test syntax structure is no longer merely a testing preference; it is a measurable software design concern that affects AI code generation quality.

*Data Availability.* All experimental data (830+ generated files, prompts, model responses), analysis scripts, the MI toolkit, and experiment runner will be made available upon acceptance.

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

**Temporary page!**

LaTeX was unable to guess the total number of pages correctly. As there was some unprocessed data that should have been added to the final page this extra page has been added to receive it.

If you rerun the document (without altering it) this surplus page will go away, because LaTeX now knows how many pages to expect for this document.