# OpenReview forum: "Co-Located Tests, Better AI Code: How Test Syntax Structure Affects Foundation Model Code Generation"
_ACM.org/AIWare/2026/Conference — AIware 2026_

### Official Review · Reviewer_Dn6X · 2026-03-06

**Rating:** 3
**Confidence:** 3

**Review:**

Strengths

•	Broad model coverage: The empirical study covers 12 mainstream models and the mechanistic analysis includes 7 models with diverse architectures.

•	Comprehensive evaluation: Three dimensional orthogonal evaluation framework, including determinism, preservation and correction.

Weaknesses

•	The empirical study is limited to a single task (d-ary heap implementation). Although the authors explain the rationality of this choice, it limits the generalizability of the conclusions to different code task domains.

•	Incomplete coverage of the test co-location spectrum: only the two extremes of the spectrum are selected, and the more mainstream cross-file separated test modes (such as pytest, JUnit) are not covered.

Comments for authors

Novelty

•	The paper takes test syntax structure and co-location degree as the core element of software design to explore its impact on AI code generation, which is different from the previous studies that ignore test preservation and structural impact. The proposed SEGA framework takes determinism, preservation and correctness as orthogonal dimensions for evaluation, and distinguishes the evaluation granularity between file level and individual test case level. Since I am not the expert in this field, I cannot accurately assess the innovativeness of this work.

Significance

•	The research clarifies the quantifiable impact of test syntax structure on the quality of AI code generation. The proposed guidelines include the test framework selection, CI/CD pipeline design and model selection of AI coding assistants.

•	The research reveals that test preservation and functional correctness are two independent evaluation dimensions, breaking the cognition of "preserved tests mean correct code", and providing empirical basis for the improvement of the quality evaluation theoretical system of AI code generation.

Soundness

•	The overall empirical research design of the paper is rigorous. The baseline experiment first verifies that all models have the ability to correctly implement the d-ary heap, eliminating the interference of the model's own coding ability on the experimental results.

•	For the experimental design and robustness of the conclusions, there are two improvement suggestions as follows:

1.	For the validity threat of single task, it is recommended to supplement small-scale verification experiments of 1-2 different types of code tasks or more fully discuss the limitations of task singleness and the supporting basis for the task independence of marker-level effects in MI analysis.

2.	For the incomplete coverage of the test co-location spectrum, it is recommended to supplement a small-scale comparative experiment on cross-file separated test mode (such as pytest, JUnit), or clearly elaborate the feasibility and potential limitations of extending the conclusions of this study to cross-file test scenarios in the discussion section.

Replicability

The paper commits to publicly releasing all experimental data (over 830 generated files, prompts, model responses), analysis scripts, MI toolkit and experiment runner upon acceptance. [ag3.1]It is recommended to supplement the open-source link in the paper, so as to improve the data accessibility of the research and also better align with the open-source norms of scholarly research.

Presentation

•	The overall structure of the paper is clear, the logic is coherent.

•	Lack of detailed text description of the experimental conditions shown in Table 1.

•	The annotation "∗Except Claude 3.5 Haiku" in Section 4.2 is recommended to be supplemented to the table note of the table 5 instead of the text, for more standardized format.

•	For the calculation method of effective attention of RWKV-6 in Section 5.1, it is recommended to supplement more detailed calculation steps to improve the replicability of the mechanism analysis part.

**Summary:**

This paper investigates how the test syntax structure affect the quality of code generated by foundation models. Through a large-scale empirical study with over 830 generated files, 12 models across 3 providers, the authors propose the SEGA three-dimensional evaluation framework measuring Determinism, Preservation, and Correctness. They conduct a systematic comparison between inline test syntax (Python doctests, maximum co-location) and separated test syntax (Rust #[test] blocks, same-file separation) on a d-ary heap implementation task. The paper further provides mechanistic evidence via attention pattern analysis, knockout experiments, and steering interventions across 7 open-source architectures (6 transformers and 1 gated-linear RNN). The core conclusion is that co-locating tests with implementation code produces measurably better AI-generated code, and the authors put forward actionable design guidelines for AI-powered development.

---

> ### Author Response · Authors · 2026-03-17
> **Single-Task Validity, Co-location Spectrum, and Presentation**
>
> Thank you for the constructive review and specific improvement suggestions.
>
> **Single-task validity**: Please see our global comment (point 2). We will strengthen the discussion of this limitation. The MI marker-level evidence and our additional cross-language results (same task, 5 languages, 10 models) suggest the effect is syntax-driven rather than task-specific.
>
> **Incomplete co-location spectrum**: We agree that cross-file separated tests (pytest, JUnit) represent an important middle point. Our study deliberately chose the two extremes to establish whether a measurable effect exists at all. Having established that it does, the natural next step is to test intermediate points on the spectrum. We will add a discussion of the feasibility and expected behavior for cross-file test modes in the revision.
>
> **Open-source link**: We will add a repository link in the revision. We commit to full release of all data, scripts, and the MI toolkit.
>
> **Table 1 description**: We will add detailed textual descriptions of experimental conditions. Thank you for flagging this gap.
>
> **Claude 3.5 Haiku annotation**: Agreed — we will move this to a table note for Table 5.
>
> **RWKV-6 effective attention calculation**: We will add detailed calculation steps to the paper or appendix. Briefly: we compute effective attention from the WKV recurrence by extracting signed *r · k* products with cumulative decay factors, applying ReLU to retain only positive contributions, then L1-normalizing per-position to produce comparable [batch, heads, seq, seq] matrices. Full implementation details will be in the released toolkit.

---

### Official Review · Reviewer_CBDh · 2026-03-10

**Rating:** 2
**Confidence:** 3

**Review:**

- My main concern is that the central comparison is heavily confounded. The paper compares Python doctests to Rust #[test] blocks, which changes language, framework, syntax, and ecosystem convention at the same time. The authors describe these as two extremes of a co-location spectrum, but that also means the study does not cleanly isolate.

- I also think the paper overstates generality. Although it reports 830+ generations, the substantive evidence is still mostly from one d-ary heap task, and the paper explicitly acknowledges this as a threat to validity. So this is a strong case study, but not yet a strong basis for a broad design principle.

- The paper’s baseline is: Python, model-generated doctests, on the d-ary heap task. The main comparison it later wants to make is against Rust, prompt-provided #[test] blocks, on the same task. In other words, the baseline and the critical condition differ on more than one axis at once: language, test syntax, and even the prompting regime (“generate your own tests” vs “preserve tests that were supplied to you”). A matched control should ideally change only the factor of interest. This one does not. So when the paper says the baseline establishes that later differences are due to “test handling, not coding ability,” that is too strong. The baseline shows that the models can solve the heap task in one favorable Python setting; it does not cleanly prove that the later Python–Rust gap is caused only by syntax/co-location rather than by broader language or setup effects.

- The mechanistic section is interesting but over-claimed. Its evidence comes from seven small open models rather than the proprietary models driving the main results; the effect is not universal across architectures; the RWKV causal evidence is only marginal; and the steering gains are initially shown on five prompts before broader scaling reveals that the small models are too weak for reliable Rust generation. I therefore view the MI results as suggestive supporting evidence rather than strong justification for claims about robustness to future architectures.


Pros

- Somewhat relevant question.
- Easy to understand and readable presentation.

Cons
- Main comparison is confounded.
- Evidence is mostly one task.
- Broad claims are stronger than the evidence.
- No specific section for RQ4. I guess Section 5 is about RQ4 where as all other questions were organized within Section 4 made it bit confusing.
- Mechanistic section is suggestive rather than decisive.
- No example of prompts

**Summary:**

The paper studies whether test syntax structure affects AI code generation quality. It organizes the study around four questions: whether inline tests improve generation quality, whether the effect is due to syntax rather than language, whether it changes across model generations, and what internal mechanism explains it. They introduced SEGA, a three-part evaluation based on determinism, preservation, and correctness; compare Python doctests with Rust #[test] blocks on a d-ary heap task across 12 proprietary models; and add mechanistic analyses on 7 smaller open models. The headline claim is that inline/co-located tests lead to near-perfect results, while separated tests expose large gaps.

---

> ### Author Response · Authors · 2026-03-17
> **Isolating Co-location from Confounds**
>
> Thank you for the direct and detailed critique. We address your specific concerns:
>
> **Confounded comparison**: Please see our global comment (point 1). We want to emphasize the tier inversion specifically: the confounding hypothesis predicts that model capability should correlate with Rust performance. Instead, Haiku 4.5 (lowest tier) achieves 100%/100% in Rust while Opus 4/4.1/4.5 (highest tier) suppress all tests despite writing correct Rust code. This pattern is difficult to explain via language difficulty or model priors, but follows directly from how each model handles separated test syntax. Additionally, our cross-language experiments (5 languages, 10 models) show that identical test code in the same language produces opposite preservation results depending solely on inline vs. import presentation — directly isolating structure from language.
>
> **Baseline does not isolate factors**: We agree the baseline alone does not prove the gap is caused only by syntax. The baseline establishes capability; the causal argument rests on the *combination* of baseline + tier inversion + MI marker-level evidence + cross-language replication. We will revise to make this reasoning chain more explicit rather than relying on the baseline claim alone.
>
> **Mechanistic section**: We agree it should be framed as suggestive supporting evidence, not decisive proof. We will revise accordingly. We note, however, that the MI analysis was conducted on open-source models by necessity (proprietary models do not expose internals), and we view the consistency across 5/7 architectures (including a non-transformer) as meaningful even if not definitive.
>
> **No RQ4 section**: Section 5 does address RQ4. We will add an explicit "RQ4" label to the Section 5 heading to improve navigation.
>
> **No prompt examples**: We will include representative prompts in an appendix or supplementary material.
>
> **Evidence is mostly one task**: Please see our global comment (point 2).
>
> We are prepared to incorporate a summary of the cross-language results in a revision, replacing or condensing other material if needed, should reviewers find this would strengthen the paper.

---

### Official Review · Reviewer_oZZz · 2026-03-11

**Rating:** 3
**Confidence:** 4

**Review:**

## Strength
+ They tackle a timely and interesting question, whether test syntax structure influences AI code generation quality, and frame it in a way that feels both practically useful and intellectually fresh.
+ The topic is relevant for AI-assisted software development, especially because it connects prompt and test design choices to broader engineering practice.
+ The paper is well organized and generally easy to follow. The progression from the empirical findings to the discussion is clear.

## Weakness
- The core empirical observation is interesting, but the paper may go a bit too far in presenting co-location itself as the established causal explanation.
- Some of the broader implications for framework design and development practice feel slightly ahead of the evidence currently shown.
- The main comparison still combines several factors at once, including language, test style, and likely model priors, so the central causal claim is not fully isolated.
The methodology is clearly presented, but some interpretive claims would benefit from more careful scoping.

## Detailed Comments

### Novelty
This paper asks a thoughtful and timely question, which its main strengths. The idea that test structure might influence model behaviour is both plausible and practically relevant, and the authors make that idea concrete in a compelling way. At the same time, I think the novelty claim would be even stronger if it were framed a little more carefully. What they clearly establish is that the specific contrast studied here, Python doctests versus Rust #[test] blocks, leads to a substantial difference in model behaviour, which I think is a valuable contribution. Where I became less convinced is in the stronger claim that co-location itself is the causal principle. Since the design does not fully separate co-location from language, ecosystem convention, or prompt-local example structure, I do not think the current version fully supports that broader interpretation. A slightly narrower framing would still leave the paper with a more convincing contribution.
### Significance

The manuscript is clearly motivated. However, some of the downstream implications feel a little broader than the present evidence base. The study uses one task and one cross-language contrast, yet the discussion extends to test framework design, IDE design, and general development practice. Those implications are certainly interesting, and some may well turn out to be correct, but they would be more convincing if supported by additional evidence across more tasks or more controlled comparisons. As written, I think the paper already makes an important practical point, namely that test presentation format can materially affect code generation quality. The stronger recommendations could be presented more as promising directions suggested by the evidence rather than as conclusions already settled.

### Soundness

This is the area where I think the paper would benefit most from revision. The central argument is that the observed difference is driven by test syntax structure rather than language difficulty, but the current design does not fully isolate those factors. The comparison is between Python doctests and Rust #[test] blocks, which differ not only in co-location but also in language, testing culture, and likely model training priors. The paper does make a reasonable effort to address this, especially by showing that models can generate correct Rust code while still failing to preserve tests. That is useful evidence, but it does not completely rule out an alternative explanation, namely that models have simply learned different continuation conventions for doctests and Rust test blocks. I also found the mechanistic section interesting and well motivated, but I would treat it as supportive rather than decisive evidence, especially since it is conducted on smaller open models rather than the same proprietary models featured in the behavioural results. Overall, I think the empirical effect is there and important, but the causal interpretation should be stated a bit more cautiously.

### Presentation

The paper is generally well written and easy to read. The structure is clear, and the tables are helpful. My main presentation concern is not clarity, but calibration. In a few places, the language sounds stronger than the results later justify. For example, the abstract and early framing suggest near-perfect inline-test performance across all models, but the body later notes an explicit exception for Claude 3.5 Haiku. This kind of mismatch is easy to fix, but it matters as it can make readers feel that the claims are slightly oversold. I had a similar feeling in parts of the discussion where the mechanistic findings are used to support broader design guidance. The underlying results are interesting, so I do not think the paper needs especially strong phrasing to make its case. A more measured tone in a few places would actually make the argument more persuasive.

### Transparency/Reproducibility

I want to emphasize that the paper does several things well here. The experimental setup is understandable, the evaluation categories are clearly defined, and the study is more transparent than many papers that make similar claims. My concern is less about methodological opacity and more about interpretive transparency. In particular, I think the paper would benefit from being more explicit about the boundary between what is directly demonstrated, what is supported but still tentative, and what remains a plausible hypothesis. The limitations section does acknowledge important caveats, which I appreciated, but some of those caveats are not fully reflected in the tone of the abstract and conclusion. Tightening that alignment would improve the paper and make its contribution feel more precise and trustworthy.

## Minor Comments

1. Abstract, p. 1: The phrase saying inline tests yield near-perfect preservation and correctness “across all models” appears inconsistent with the later note that Claude 3.5 Haiku strips all doctests.
2. Contributions / early framing, p. 1: Similar wording about “across all tested models” should be qualified for consistency.
3. §4.2, p. 3: The heading “Near-Perfect AI Performance” feels slightly too strong given the stated exception.
4. Throughout: The terminology for the non-transformer model family could be made a bit more consistent.

**Summary:**

The paper shows that the way tests are written in a prompt materially affects AI code generation quality: when tests are co-located inline with the implementation, such as Python doctests, models preserve the tests and generate correct code far more reliably than when tests are separated into distinct blocks, such as Rust #[test] functions. Across 830+ generated files from 12 models, the authors find near-perfect preservation and high correctness for inline tests, but large model-tier differences and frequent test suppression for separated tests, and they further support this with mechanistic analysis showing that inline test markers are more strongly tied to the relevant code inside the model. The main takeaway is that test syntax structure is not just a stylistic choice in AI-assisted development; it is a practical software design decision that can improve the quality and trustworthiness of generated code.

---

> ### Author Response · Authors · 2026-03-17
> **Calibrating Claims and Causal Framing**
>
> Thank you for the nuanced and well-calibrated review. Your distinction between "what is directly demonstrated, what is supported but tentative, and what remains a plausible hypothesis" is exactly the framing we should adopt.
>
> **Co-location as causal explanation**: We agree the current framing goes too far. We will revise to present the empirical finding (inline tests produce measurably better results than separated tests in this study) as the established contribution, and co-location as the best-supported mechanistic explanation rather than a fully isolated cause. As noted in our global comment, additional cross-language evidence — where identical test code in the same language produces opposite results depending on inline vs. import presentation — further supports this interpretation, but we acknowledge that alternative explanations (learned continuation conventions) are not fully ruled out.
>
> **Implications ahead of evidence**: We accept this. The design guidelines will be reframed as "directions suggested by the evidence" for the broader recommendations (framework design, IDE design), while retaining the more direct practical points (run your tests, evaluate per-model) as conclusions.
>
> **Calibration of language**: We will implement all the specific changes you suggest:
> - Abstract: qualify "across all models" with the Haiku 3.5 exception
> - Section 4.2 heading: adopt a more measured title
> - Discussion: clearly distinguish demonstrated findings from supported-but-tentative implications
> - Align abstract/conclusion tone with the body's nuances
>
> **Terminology consistency for non-transformer models**: We will standardize on "gated-linear RNN" throughout.
>
> Your observation that "a more measured tone would actually make the argument more persuasive" resonates with us and will guide the revision.

---

> > ### Comment · Reviewer_oZZz · 2026-03-20
> >
> > Thank you for the response. The points on claim calibration and single-task scope are adequately addressed for me, and I appreciate the planned revisions there.
> >
> > The only point that still feels not fully resolved is the causal interpretation. The added evidence, especially the tier inversion, corrects Rust with suppressed tests, and the marker-level MI analysis, does strengthen the case that this is not simply about Rust difficulty. The cross-language companion study is also relevant and moves in the right direction.
> >
> > What still keeps this only partially addressed is that the strongest new evidence, the companion study, is not part of the paper itself and is only summarized briefly in the rebuttal, so it is hard to assess its strength from the response alone. More importantly, even with the added detail, the alternative explanation in terms of learned model priors or conventions around different test presentations still seems to remain open, as you also acknowledge. So I now feel more comfortable with a carefully framed version of the claim, but not with co-location as a fully isolated cause.
> >
> > Overall, this improves my confidence, but my original assessment was already positive enough, so I am not changing my score. I remain at Weak Accept.

---

### Author Response · Authors · 2026-03-17
**Addressing Shared Concerns (Confounding, Generalizability, Claim Calibration)**

We thank all three reviewers for their thoughtful and constructive feedback. We address the three shared concerns below, then respond individually to each review.

### 1. On confounding (all reviewers)

We agree that the Python-vs-Rust comparison changes multiple factors simultaneously, and we appreciate this being flagged clearly. We want to highlight three pieces of evidence *already in the paper* that isolate test syntax structure as the primary factor (a), (b), (c), plus additional cross-language evidence (d):

(a) **Tier inversion** (Section 4.3): If language difficulty explained the gap, higher-tier models should perform *better* on Rust, not worse. Instead, Haiku 4.5 (lowest tier) achieves 100% preservation and 100% correctness in Rust, while Opus 4/4.1/4.5 (highest tier) suppress all tests. This inversion is inconsistent with a language-difficulty explanation but entirely consistent with a test-syntax-handling explanation.

(b) **Correct Rust, suppressed tests** (Table 5): Opus 4/4.1/4.5 generate correct, compiling Rust implementations — they demonstrably *can* write Rust. They suppress the `#[test]` blocks specifically. The failure is in test syntax handling, not language capability.

(c) **MI at marker level** (Section 5.1): The attention analysis measures binding strength of test *markers* (`>>>` vs. `#[test]`) to function tokens, not language-level features. The 2.8–4.4× difference operates at the syntactic marker level.

(d) **Additional cross-language evidence** (omitted for space): We had conducted a companion study across 5 languages (Go, Rust, C++, TypeScript, Zig) and 10 models, but it was omitted from the current paper due to page constraints; it is available to reviewers upon request. In that study, identical test code in the *same language* produces 0% or 100% test preservation depending solely on whether tests are presented inline or via an import reference — directly isolating structural presentation from language. Python doctests achieved 100% preservation across all 10 models; Rust and Zig inline tests achieved 100% with Sonnet-class models; external-test languages (Go, C++, TypeScript) achieved 0%.

We acknowledge this does not fully eliminate all possible confounds (e.g., model training priors for different test conventions), and we will revise the paper to state the causal claim more carefully, framing co-location as the best-supported explanation rather than a fully isolated cause.

### 2. On single-task generalizability (all reviewers)

We acknowledge that the d-ary heap is a single task and that this limits generalizability. The choice was a deliberate depth-over-breadth tradeoff driven by two constraints: (a) cost (*n*=50 runs × 12 models × multiple experiments), and (b) the MI analysis requires a task at the capability boundary of 3B–7B models — simpler tasks would be trivial and uninformative, while harder tasks would exceed these models' abilities entirely.

Two factors suggest the finding is not task-specific: (i) the MI evidence operates at the test *marker* level, not the task-content level — the attention binding of `>>>` to `def` does not depend on whether the function implements a heap or a sorting algorithm; (ii) the cross-language experiments mentioned above replicate the structural effect on the same task across 5 languages and 10 models, confirming the mechanism is syntax-driven. We will strengthen the discussion of this limitation and its mitigation in the revision, and we note that multi-task validation is a clear direction for future work.

### 3. On claim calibration (Reviewers CBDh, oZZz)

We accept this feedback. The abstract and framing use language that is stronger than the evidence warrants in several places. We will revise to:
- Change "across all models" to "across all but one model" (or qualify with the Claude 3.5 Haiku exception)
- Retitle Section 4.2 from "Near-Perfect AI Performance" to a more measured heading
- Frame the design guidelines as "evidence-supported recommendations" rather than established conclusions
- Consistently present the MI section as "supporting evidence" throughout
- Align the abstract and conclusion tone with the nuances acknowledged in the body